# 100 Hz ROCS microscopy correlated with fluorescence reveals cellular dynamics on different spatiotemporal scales

Felix Jünger[1], Dominic Ruh[1], Dominik Strobel[1], Rebecca Michiels[1], Dominik Huber[1], Annette Brandel[2,3], Josef Madl[3,4,5], Alina Gavrilov[6], Michael Mihlan [6], Caterina Cora Daller [4], Eva A. Rog-Zielinska[4,5], Winfried Römer [2,3,7], Tim Lämmermann[6] & Alexander Rohrbach [1,2,3,5✉]

Fluorescence techniques dominate the field of live-cell microscopy, but bleaching and motion blur from too long integration times limit dynamic investigations of small objects. High contrast, label-free life-cell imaging of thousands of acquisitions at 160 nm resolution and 100 Hz is possible by Rotating Coherent Scattering (ROCS) microscopy, where intensity speckle patterns from all azimuthal illumination directions are added up within 10 ms. In combination with fluorescence, we demonstrate the performance of improved Total Internal Reflection (TIR)-ROCS with variable illumination including timescale decomposition and activity mapping at five different examples: millisecond reorganization of macrophage actin cortex structures, fast degranulation and pore opening in mast cells, nanotube dynamics between cardiomyocytes and fibroblasts, thermal noise driven binding behavior of virus-sized particles at cells, and, bacterial lectin dynamics at the cortex of lung cells. Using analysis methods we present here, we decipher how motion blur hides cellular structures and how slow structure motions cover decisive fast motions.

[1] Laboratory for Bio- and Nano-Photonics, Department of Microsystems Engineering - IMTEK, University of Freiburg, 79110 Freiburg, Germany. [2] Faculty of Biology, University of Freiburg, Schänzlestraße 1, 79104 Freiburg, Germany. [3] Signalling Research Centres BIOSS and CIBSS, University of Freiburg, Schänzlestraße 18, 79104 Freiburg, Germany. [4] Institute for Experimental Cardiovascular Medicine, University Heart Center Freiburg - Bad Krozingen, and Faculty of Medicine, University of Freiburg, Elsässer Straße 2q, 79110 Freiburg, Germany. [5] German Collaborative Research Centre SFB1425, University of Freiburg, Freiburg, Germany. [6] Max Planck Institute of Immunobiology and Epigenetics, Stübeweg 51, 79108 Freiburg, Germany. [7] Freiburg Institute for Advanced Studies (FRIAS), University of Freiburg, 79104 Freiburg, Germany. ✉email: rohrbach@imtek.de

Living cells consist of a large variety of proteins and other molecules, undergoing thermal and transported motions as monomers, filaments, membranes, and globular structures. All these cellular structures move the faster, the smaller they are. However, the smaller the cross-section of a structure, the less light will it typically re-emit by fluorescence or by direct (coherent) light scattering after illumination. And this is a significant problem in life cell microscopy: recording the motion of a dynamic structure requires a sufficiently high number of photons on the camera to enable short integration times[1–4], thereby avoiding motion blur. But the problem is even more severe, since the signals of moving structures can disappear completely in the background noise, if the signal intensity is spread out over too many pixels in a given time window. Hence, a meaningful image requires enough photons detected per pixel and time window, which are usually not available for fast and small structures.

For example, a 100 nm spherical particle diffuses over an average distance of $x_{rms} = 200$ nm within only 10 ms (in a medium 5x more viscous than water). Within 100 ms camera integration time, it diffuses already $x_{rms} \approx 600$ nm, making a particle practically invisible because of significant motion blur. And in water the situation would be 5x more severe.

Another problem in life cell microscopy is that often the decisive moments for the biological interactions cannot be predicted. Sometimes thousands of images need to be recorded, if the decisive interactions take place on the millisecond timescale.

Despite the well-accepted strengths of fluorescence imaging techniques, fluorophore bleaching and the limited output of fluorescence photons restrict many investigations in dynamic living cells. Laser light, which is known for its strong (spatial and temporal) coherence and interference ability, shows a way how to achieve both high temporal and high spatial image resolution. Interference means that a photon scattered at a structure can be amplified in its number (or intensity), when its electric field $E_1$ is multiplied with another, stronger electric field $E_2$, such that the amplification is $E_1 \cdot E_2$. Interferometric enhancement was often difficult to apply in cell biology, because of too many unwanted interference patterns (laser speckles) arising from the strongly structured surfaces of cells.

Increasing spatial resolution requires more complex illumination patterns and more illumination time. Resolution doubling can be achieved by oblique illumination, leading to either a modulation of the electric field in the coherent case or a modulation of the intensity in the fluorescence case. In the latter case, two oblique laser beams must interfere as in the case of structured illumination microscopy[3,5,6]. Oblique illumination allows to access information from light diffracted under large angles effectively increasing the detection numerical aperture $NA_{det}$.

Rotating coherent scattering (ROCS) microscopy is an emerging technique, able to overcome the above-mentioned drawbacks: first, by increasing spatial resolution nearly twofold through highly oblique sample illumination, and second, by rotating a collimated laser beam over a $2\pi$ azimuthal angle leading to nearly speckle- free images. These static and deterministic speckles are nothing else but deformed images of object structures, which locally shift the phase and result in multiple interferences on the camera. Speckles are induced by the small objects structures and therefore must not be averaged out, but assigned correctly to the object, which is achieved by the ROCS rotation. Hence, by the $2\pi$ angular integration, the deformed multiple interferences (speckles) reform into meaningful, clear images of the objects[7,8].

Similar to fluorescence techniques, evanescent sample illumination in total internal reflection (TIR) mode removes unwanted background signals and increases image contrast also in ROCS microscopy.

Other label-free contrast-enhancing techniques, such as differential interference (DIC) microscopy, phase contrast microscopy or reflective wave interference microscopy, are limited in spatial resolution by the minimal spot distance of half the used light wavelength λ. This limits resolution also in iSCAT microscopy, which can measure the smallest refractive index changes, but is not well suited for cell imaging[9]. Conventional resolution at λ/2 is achieved by gradient light interference microscopy (GLIM), which allows 3D imaging of refractive index changes[10]. Optical diffraction tomography[11,12], which is able to reconstruct phase objects in 3D, uses an oblique rotating beam as well, which is typical for tomography applications, but does not rely on speckle interferences to maximize the contrast of small structures and has not yet achieved spatial resolutions below 200 nm[13].

In comparison to our earlier study[14], we have improved the ROCS technology by nearly doubling the mean acquisition rate, by varying the polar illumination angle and the laser wavelength, thereby adapting the illumination scheme to the individual cells and adhesion properties, which leads to significantly increased image quality. Furthermore, we are now able to record ROCS signals and fluorescence signals at the same time, thereby combining the specificity of fluorescence imaging with the millisecond dynamics of label-free ROCS imaging. Although structure-specific light emission is currently not possible with ROCS microscopy, we extract relevant biological information even without specificity that cannot be obtained by any other imaging technology.

In the current study, we investigate the relevance of spatio-temporal resolution, where we demonstrate that limited temporal resolution strongly affects the spatial resolution due to motion blur. We introduce several methods to analyse stacks of $10^2$–$10^4$ images obtained by 100 Hz ROCS (and 1–10 Hz fluorescence) imaging, such as frequency decomposition, where cellular activity can be analyzed and compared on different timescales. This methodological combination provides a clearer picture of how virus-sized particles bind tighter to cellular binding sites (receptors) over time, how filopodia tips undergo a so far undetected search-and-tumble mechanism, how actin backbones undergo ultrafast retrograde flow upon addition of Latrunculin, how extracellular tunneling nanotubes of cardiac fibroblasts (connecting to cardiomyocytes) vibrate at 50–100 Hz and become stiffer over time, how mast cells open degranulation pores in times <10 ms (requiring additional hypotheses about cytoskeleton reorganization and force generation) and how bacterial lectin A patches move with enormous velocities within and underneath the cell membrane, requiring unexpected actin densities and myosin transport mechanisms.

## Results

**Principles of ROCS microscopy.** We have developed an improved ROCS microscope, which is able to use three different laser beams (at 405 nm, 445 nm, and 561 nm) at different polar angles for variable ROCS imaging depending on the cells properties and their adhesion to the coverslips. ROCS can be acquired in parallel to fluorescence imaging with either the same laser wavelength or a different laser line. In addition, incoherent images can be taken with a blue LED (at 435 nm) for control and comparisons. As sketched in Fig. 1a, three different laser beams rotating at 100 Hz by galvanometric scanners are focused in variable diameters into the back focal plane of the objective lens (HCX PL Apo, NA 1.46, Leica), to enable oblique sample illumination at variable polar angles θ. For each azimuthal angle φ, the plane (or the evanescent) wave is scattered at cellular structures, where the backscattered light is captured by a CMOS camera (PointGrey GS3- U3-23S6M-C, Richmond BC, Canada),

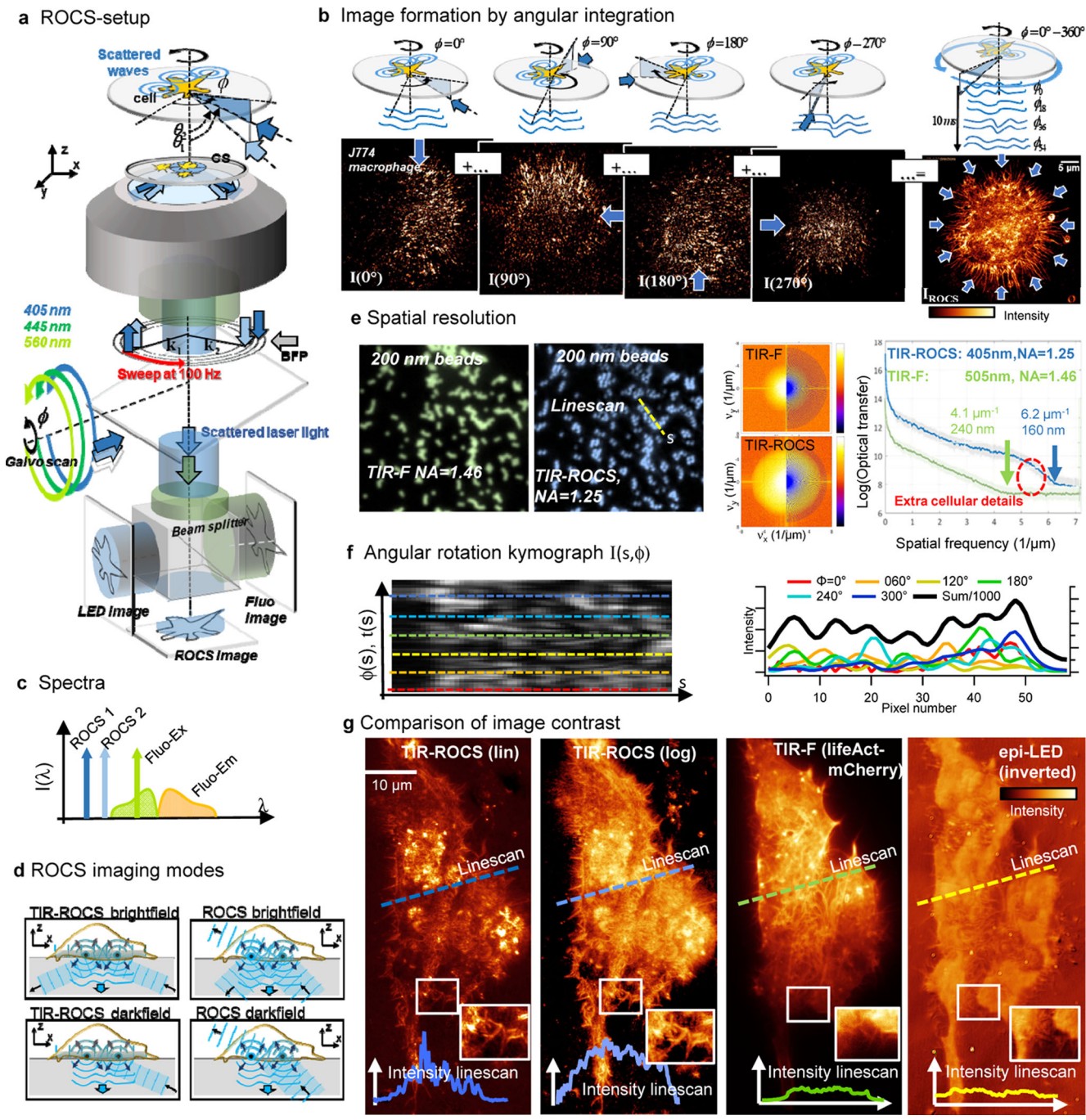

**Fig. 1 Principles of ROCS microscopy. a** Setup scheme of ROCS microscope with azimuthal laser beam rotation at 100 Hz incident under high polar angles θ. Scattered laser light is imaged onto a camera integrating over sweep time. **b** ROCS images form upon the full 360° sweep of a laser beam. Coherent images from different azimuthal directions (see blue arrows) form speckle patterns, which generate the final ROCS image by incoherent superposition. **c** Wavelength spectra I(λ) for ROCS illumination and detection at the same wavelength (405 nm or 445 nm), without exciting fluorescence. Fluorescence excitation with laser (560 nm), fluorescence emission at orange wavelengths. **d** Four possible illumination and detection modes in ROCS microscopy, where TIR-ROCS in darkfield mode provides the highest contrast. **e** Parallel TIR-imaging of 200 nm PS beads in fluorescence mode (green at 505 nm, at NA = 1.46) and ROCS mode (blue at 405 nm, NA = 1.2). Spatial frequency spectra for TIR-F and TIR-ROCS are averaged in azimuthal direction (logarithmic intensity, STD in light gray), shown by green and blue line scans. TIR-F provides 240 nm spatial resolution, TIR-ROCS achieves 160 nm. **f** Kymograph I(s,ϕ) along direction s (dashed line in ROCS image). For totally n = 72 raw images from different azimuthal angles ϕ the speckle patterns show limited angular correlations. Below: Six line scans I(s,ϕ_n) at six angles with resulting sum intensity. **g** Comparison of image contrast for different imaging modes of a HeLa cell. Left to right: TIR-ROCS image (linear intensity) recorded within 10 ms at 445 nm with large dynamic range. The logarithmic intensity image shows lower intensity structures (filopodia) with strong signal contrast. The TIR-F image (200 ms integration time) with Lifeact-mCherry actin labeling reveals similar, but not the same structures (see ROI). An incoherent epi-image (inverted) of a blue LED (435 nm) reveals again different cellular details at low contrast and dynamic range (purple line). Source data are provided as a Source Data file.

acquiring images at 100 Hz. A complete azimuthal 360° scan is performed within 10 ms, such that scattering intensity patterns (speckle patterns) from different illumination angles are added up within the integration time of the camera $\tau = (100 \text{ Hz})^{-1}$. For more technical details see Supplementary Note 1.

The image formation process is explained by Fig. 1b, where the light scattering from four exemplary illumination angles ($\phi_0 = 0°, 90°, 180°, 270°$) is sketched, together with the coherent wave trains propagating towards the camera. The exemplary intensity images $I(\phi)$ formed by each wave train (see blue distorted wavefronts) are visible as speckle patterns below, which do not resemble the image of a cell. However, after adding up e.g., 72 images from 72 equally distributed illumination directions, a high contrast, superiour-resolved image of a J774 mouse macrophage becomes visible (see Supplementary Movie 1). Notably, only for Fig. 1b, we have recorded 72 individual images to superimpose them. Usually, the angular intensity summation is performed during the 10 ms integration time of the camera. Due to the scanning process, a ROCS image is spatially coherent in the radial direction (summation of field amplitudes) and incoherent in the azimuthal direction (summation of intensities).

Figure 1c illustrates that the spectra for ROCS imaging are very narrow and are not shifted. Typically, 80% of the illumination light and 20% of the detection light are lost at the beam splitter, which is dichroic for parallel fluorescence imaging. It is possible to use one single laser for ROCS imaging and fluorescence excitation at the same time and to detect the Stokes-shifted fluorescence light on a second camera (Hamamatsu ORCA Flash V2 4.0). However, to avoid bleaching by continuous ROCS illumination at typically 1 mW laser power (at 405 nm or 445 nm) at the sample, we used a rotating, time-switching 560 nm laser beam for fluorescence excitation of RFP (red fluorescent protein) or mCherry-labeled structures (emission maximum at 610 nm), which hardly absorb at 405 nm or 445 nm. To set up an efficient ROCS microscope is of outmost importance to minimize the reflected light ($R < 0.1\%$) at all interfaces of lenses and filters, since this light can become easily brighter than the light backscattered from small sample structures.

ROCS can be used in four different imaging modes (Fig. 1d) depending on the ratio of the illumination numerical aperture $NA_{ill}$ and the detection $NA_{det}$. For cases $NA_{det} < 0.85 \cdot NA_{ill}$, ROCS can be used in darkfield (DF) mode, leading to high image contrast, when the illumination light is blocked by a diaphragm. The case $NA_{det} > 0.85 \cdot NA_{ill}$ corresponds to brightfield (BF) mode. If in addition $NA_{ill} > n_{med}$, (refractive index $n_{med}$ of cell medium), the TIR mode is possible both with darkfield and brightfield (see Supplementary Note 2). In most of the cases in this study we use the TIR-ROCS DF mode, which can be well compared to the TIR-F mode in fluorescence imaging because both show a dark background.

Whereas spatial resolution is decreased by 20% in TIR-F imaging by the inherent Stokes shift (e.g. 505 nm fluorescence emission, relative to 405 nm ROCS emission), TIR-ROCS imaging is decreased in resolution by around 20% because of the darkfield mode, leading to an effective $NA_{det} = 1.25$ instead of $NA_{det} = 1.46$. Nevertheless, the spatial resolution of TIR-ROCS microscopy is significantly better than that of TIR-F as illustrated by Fig. 1e. This is clearly visible by the images of fluorescently labeled 200 nm beads (FluoroBeads Yellow-Green, Thermo-Fisher), which can be easily separated by TIR-ROCS, but, as expected, not by TIR-F despite $NA_{det} = 1.46$. We estimated the spatial resolution by the maximal spatial frequency in each image spectrum for 10% above noise, indicating a realistic cutoff frequency (the spatial frequency profiles). The effective maximum optical transfer is $6.2 \, \mu m^{-1}$, corresponding to $\Delta x_{ROCS} = 160 \, \text{nm}$ resolution in ROCS microscopy, but is only $4.1 \, \mu m^{-1}$ in

fluorescence microscopy corresponding to $\Delta x_{FLUO} = 240 \, \text{nm}$ ($0.61 \cdot \lambda/NA_{det} = 220 \, \text{nm} < \Delta x_{FLUO}$). It is well known that the (incoherent) resolution estimate $0.61 \cdot \lambda/NA$ for a high-aperture lens is a strong idealization. Although the maximal bandwidth transferred remains the same, the object spectrum is shifted in ROCS, such that significantly more information is transferred close to the cutoff frequency. In Supplementary Fig. S3 we show how TIR-ROCS resolves adjacent 150 nm beads.

How fast do the speckles change with the azimuthal illumination directions? The intensity distribution $I(s,\phi)$ in kymograph Fig. 1f (along yellow dashed line $s$ in ROCS image above) describes the speckle dependence on the azimuthal angle $\phi$ or as $I(s,t)$ on the rotation time t. The length of the specular distributions along the horizontal axis is an estimation about the angular memory effect of ROCS imaging. Six exemplary vertical linescans display six intensity speckle distributions, which form the final (black) image profile after 72 line profiles have been added up. At this point it should be emphasized again, that the speckles are not averaged out, but represent a deformed interference image of an object structure. Only by superposition of all deformed interference images, the final superiour-resolution image of the structure is obtained. It was shown recently[8] that a reduction of the degree of spatial coherence led to a decrease in image resolution because of less interferences, i.e. less pronounced speckles. The role of spatial and temporal coherence is further illustrated in Supplementary Note 4.

The degree of spatial and temporal resolution can be described by the spatiotemporal bandwidth STBW[15], which is obtained by dividing spatial resolution through temporal resolution. In the case of ROCS we find a $STBW = 100 \, \text{Hz}/(0.16 \, \mu m) = 625 \, \text{Hz}/\mu m$, being e.g. 4x higher than the $STBW = 20 \, \text{Hz}/(0.13 \, \mu m)$ of fluorescence SIM (at $NA = 1.46$).

Figure 1g gives a comparison of images using the same HeLa cell (labeled with Lifeact-mCherry) and the same false color table. A linear representation of a TIR-ROCS image (10 ms integration) leaves small structures relatively dark, which can be easily made visible by a logarithmic image, exploiting the huge dynamic range of ROCS images. Although the integration time of the TIR-F image was 20 times longer (200 ms), the contrast is more than 10 times smaller. Although many details and the overall cell shape are similar both in the TIR-ROCS and in the TIR-F image, it can be easily seen that the strongest scattering structures in the cell are not those with the strongest actin staining. Interestingly, the (inverted) low-coherence, epi-illumination image of a blue LED reveals once again other details of the cell, but with a rather low image contrast (see line scans). See Supplementary Note 5 for further cell images with LED illumination and laser illumination under different polar angles (including iScat). A region of interest (white box) illustrates the related, but still different information content revealed with different imaging modes. However, since both fluorescence and ROCS are used in TIR mode, it is likely that mainly the cortex is illuminated and imaged (see Fig. 3 section).

**ROCS imaging of high cellular dynamics**. The pronounced resolution and contrast provided by TIR-ROCS at only 10 ms integration time is demonstrated in Fig. 2a with myeloid immune cells that were freshly isolated from mouse bone marrow. The multilobulated nuclei of neutrophils are visible with comparable quality as with electron microscopy[16] (shown in gray colors). Although these cells were not activated by any stimuli, we recorded high intracellular dynamics, which are displayed in the kymograph (and in Supplementary Movies 2 and 3).

Figure 2b demonstrates how motion blur from limited temporal resolution can reduce the spatial resolution. By TIR-

## a ROCS-imaging of myeloid cells at 100 Hz

Kymograph

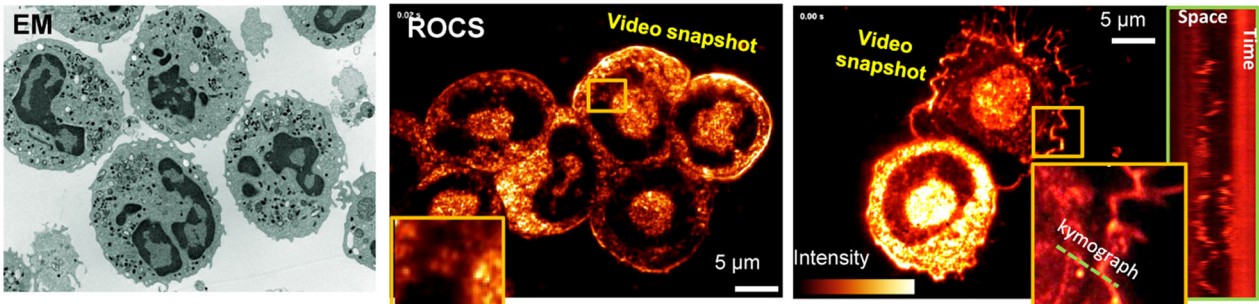

## b ROCS acquisition rates and motion blur

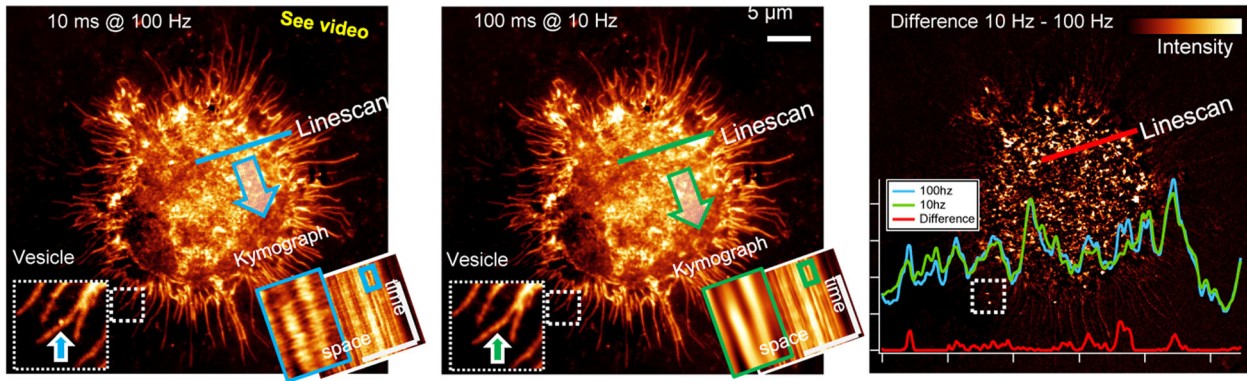

## c Displacements in cortex by optical tweezers

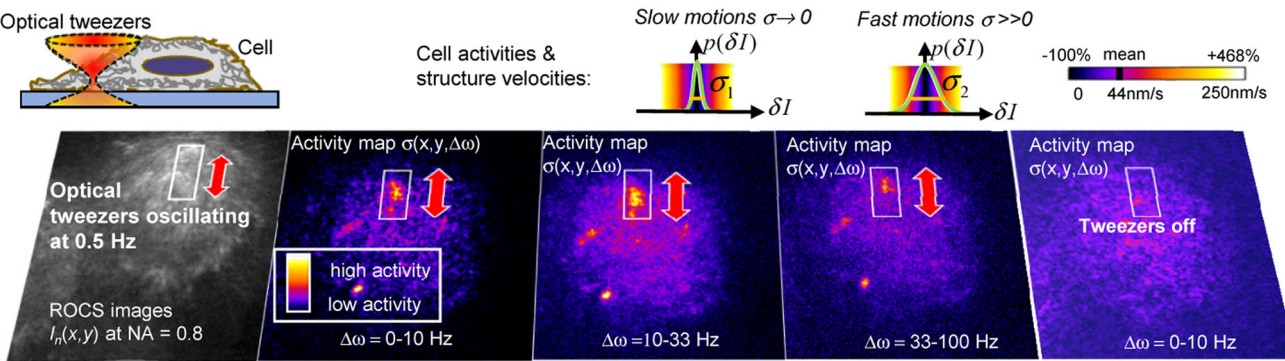

**Fig. 2 Dynamic imaging and activity maps. a** Electron micrograph of fixed myeloid cells (iStock.com/Dlumen) versus ROCS image of living myeloid cells recorded at 100 Hz. The kymograph shows cell structures moving at 100 Hz. **i** J774 macrophages image*d* with ROCS at 100 Hz (and down-sampled to 10 Hz). T*he* ROI reveals a fast vesicle at 100 Hz, which is invisible at 10 Hz due to motion blur. Intensity kymographs from line scan*s* inside the cell body reveal much higher dynamics at 100 Hz. Right: Snapshot of intensity difference between images at 10 Hz and 100 Hz with line scans in corresponding colors. **c** Oscillating optical tweezers (100 mW at 1064 nm) induce activity in the cell cortex of J774 macrophages. Non-TIR darkfield ROCS-image sequence $I_n(x,y)$ recorded at NA = 0.8 hardly reveal any cellular responses left to the red arrow. Activity maps $\sigma(x, y, \Delta\omega)$ reveal cortex activity at 3 different timescales ($\Delta\omega^{-1} = 0.1$–$1$ s, 0.03–0.1 s, 10–30 ms) within ROI (white box). Slow motions correspond to small $\sigma$ (black) and fast motions to a large $\sigma$ (yellow). Switching the optical tweezers off stops cortex activity immediately. Source data are provided as a Source Data file.

ROCS imaging a J774 macrophage at 100 Hz and by post-processive down-sampling the same image stack to 10 Hz (each 10 frames are averaged). Whereas most of the adherent filopodia are static, much higher dynamics in the cell cortex is visible in the 100 Hz movie (see Supplementary Movie 4). For a single frame, the difference between the images with 10 ms and 100 ms integration time is visible on the right together with three linescans through a central part of the cell cortex. The kymographs corresponding to these linescans and the ROI magnifications therein show much more details for the 100 Hz case. As discussed in the next section, actin reorganization is the most likely explanation for the higher activity, although single

filaments can hardly be revealed in the dense J774 macrophages. The magnification of the dashed line ROI shows the high dynamic range of ROCS imaging, but also illustrates the necessity of high temporal resolution, to reveal e.g., the existence of small fast vesicles (blue arrow). In the case of 100 ms integration time, the vesicle is invisible because of motion blur (green arrow).

**Activity mapping**. Proteins, vesicles or filaments are driven by thermal and molecular forces, hence moving differently fast on different timescales. For instance, an unknown structure moving once back and forth within 0.1 sec generates a washed-out,

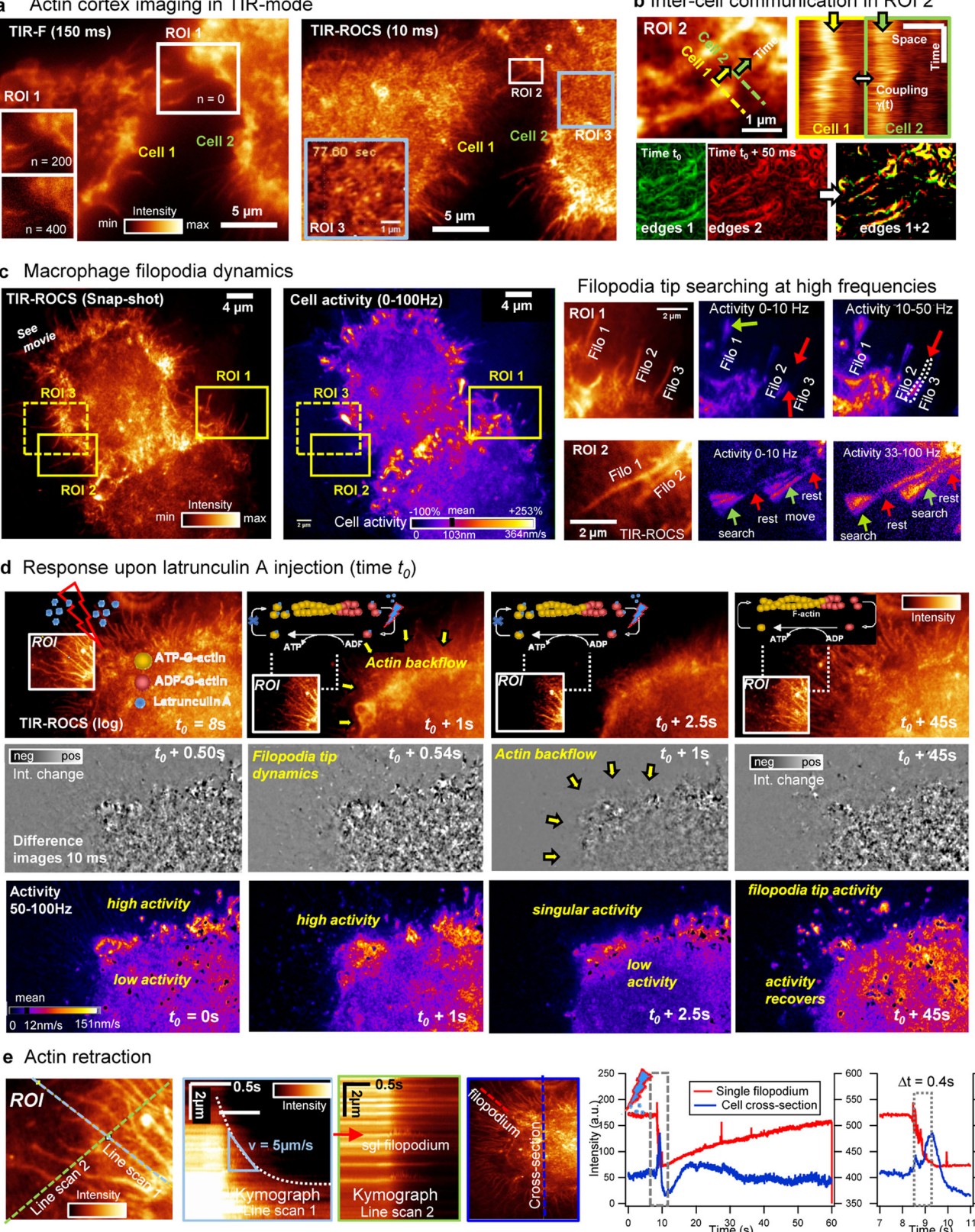

**a** Actin cortex imaging in TIR-mode

**b** Inter-cell communication in ROI 2

**c** Macrophage filopodia dynamics

Filopodia tip searching at high frequencies

**d** Response upon latrunculin A injection (time $t_0$)

**e** Actin retraction

blurred image over 0.1 s integration time, implying a broad and static structure. At 100 Hz sampling, however, one may see up to 10 different positions indicating that the structure is dynamic and is of smaller shape. Therefore, a frequency decomposition is useful when analyzing image sequences[15]. We consider typically

four different frequency ranges: $\Delta\omega_0 = 0$–$100$ Hz for a mixture of all frequencies (no decomposition), $\Delta\omega_1 = 0$–$10$ Hz for slow structure motions, $\Delta\omega_2 = 10$–$33$ Hz for fast and $\Delta\omega_3 = 33$–$100$ Hz for very fast structure motions. So-called activity maps $\sigma(x, y, \Delta\omega)$ (see Fig. 2c) are generated by analyzing 10 ms intensity

**Fig. 3 Cell cortex imaging. a** Actin cortex imaging of two MK6 macrophages (cell 1 & 2) in TIR-F mode (actin, Lifeact-GFP) at 7 Hz and in TIR-ROCS mode at 100 Hz. In TIR-F, the contact region ROI 1 reveals significant bleaching within frames n = 0, 200, 400. In TIR-ROCS, ROI 2 and ROI 3 reveal actin cytoskeleton-like structures (Supplementary Movie 6). **b** The magnified snapshot of ROI 2 marks two linescans within < 1 μm distance from cell 1 (yellow) and cell 2 (green). The two kymographs over 600 images show inter-cell communication through the cell protrusions. Two subsequent edge enhanced images of ROI 2 in green and red reveal high dynamics within Δt = 50 ms. **c** Milliseconds filopodia dynamics of J774 macrophages: 10 ms TIR-ROCS snapshot and activity plot over the complete frequency range (0–100 Hz) with three ROI. Filopodia activity is enhanced by 250% relative to mean cell activity (velocity standard deviation) of 364 nm/s. ROI 1 and ROI 2 reveal strong lateral motions in filopodia tips (green arrows), but no activity in a distance of 1–2 μm away from the tip. Little activity in frequency band Δω = 0–10 Hz, high activity in Δω = 10–50 Hz or Δω = 33–100 Hz. **d** Latrunculin A injection at time $t_0$ inhibiting F-actin polymerization especially at cell periphery. TIR-ROCS images $\log(I_n)$ at 100 Hz with ROI, 10 ms difference images $\delta I_n$ and high-frequency activity plots (0.5 s STD) reveal high cell dynamics with actin retrograde flow. After 1 s, the cell retracts, loses focal adhesions and retracts filopodia. After 2.5 s activity is minimized. After 45 s actin polymerization restarts, leading to cell extraction and filopodia growth through tip searching. **e** Magnification of ROI with green and blue line scan. Kymograph of radial linescan 1 shows high actin backflow along a single filopodium with velocities up to v = 5 μm/s. Kymograph of lateral linescan 2 shows fast actin backflow. Blue linescan (length Δy) marks cross-section through the whole cell. Intensity profiles $I_{ROCS}(t,s_0)$ of single filopodium (in red) and of whole-cell $I_{ROCS}(t,\Delta y)$ (in blue) reveal signal decrease within 0.5 s first in filopodia and 0.4 s later in the cell cortex, followed by a signal recovery within 60 s. Source data are provided as a Source Data file.

differences $\delta I_n = I_n - I_{n-1}$, 20 ms differences $\delta I_{2n} = I_n - I_{n-2}$,.., or $j \cdot 10$ ms differences $\delta I_{jn} = I_n - I_{n-j}$ of subsequent images numbered with $n$. By determining the standard deviations $\sigma(x, y, \Delta\omega)$ of these difference images $\delta I_{jn}(x, y, \Delta\omega_j)$ for each pixel (x,y), one obtains the space resolved average intensity changes within a certain time window (typically Δt = 0.1 s or Δt = 1 s). Assuming lateral motions and considering a pixel width of dx = 37 nm, one obtains standard deviations of dx per time dt, i.e. of structure velocities (either relative in % to the mean velocity or absolute in nm/s). In summary, this method describes a form of cell activity within a certain time window Δt at a specific position and on a specific timescale $\tau = 1/\Delta\omega$.

Figure 2c describes an experiment, where we have actively induced translation forces with strong optical tweezers oscillating at 0.5 Hz over 4 μm indicated by the red double arrow. The optical tweezers' laser focus (1064 nm wavelength at 100 mW) was acting underneath the cell membrane of a living J774 macrophage, which was recorded by DF-ROCS at 100 Hz (NA = 0.8). By applying an activity analysis with frequency decomposition, we could see strongly enhanced activity in the region of force application within the region of interest (white rectangle). On all timescales $\tau_1 = 100–1000$ ms, $\tau_2 = 30–100$ ms, and $\tau_3 = 10–30$ ms, we could see activities >300% higher than the mean cell activity, indicating lateral motions of cortex structures, which were likely actin filaments. The actin network is momentarily displaced by the tweezers, but relaxes first quickly and then more slowly like most viscoelastic materials, leading to activity signals $\sigma(\Delta\omega)$ on different short and long timescales $\Delta\omega^{-1}$. In ROCS, flat membranes with a constant refractive index hardly scatter light[14], whereas vesicles give strong signals. As soon as the laser tweezers and thereby lateral drag forces are switched off, the activity disappears.

**Cell cortex imaging and activity maps.** Using evanescent (TIR) illumination, we imaged the cortex of two adjacent MK6 macrophages (cell 1 and cell 2) as shown in Fig. 3a. We compare TIR-F images of Lifeact actin-GFP labeled cells (150 ms integration time at 7 Hz) with TIR-ROCS images (10 ms at 100 Hz), magnifying three regions of interest (see Supplementary Movies 5,6 and 7). The contact region between both cells contains dynamic protrusions, which appear blurry in fluorescence and become noisier with increasing number of frames (see ROIs with n = 0, 200 and 400). In the ROCS image, we magnified the smaller contact region ROI 2, which is further considered in Fig. 3b. ROI 3 magnifies highly dynamic cortex structures, possibly from the actin cytoskeleton. Due to the high dynamic range of ROCS images, the protrusions from cell 1 contacting cell 2

reveal high contrast also in the 3 μm × 4 μm ROI 2 of Fig. 3b. Two kymographs from two parallel linescans (cell 1 in yellow and cell 2 in green) in a distance of 600 nm reveal 20 Hz protrusion dynamics, with partially parallel movements, implicating motion coupling γ(t). Two subsequent, edge-enhanced frames (in green and red) show the outlines of the protrusions, revealing structural changes within 50 ms by the nonyellow regions in the overlap image.

Macrophage filopodia have diameters of typically 100–200 nm enabling even faster dynamics than the protrusions in Fig. 3b. Two other J774 macrophages are shown in figure Fig. 3c by a ROCS image (together with Supplementary Movie 8) and by the corresponding activity map (full 100 Hz bandwidth, analyzed over Δt = 0.5 s). Three regions of interest (ROI) of highly dynamic filopodia are available as Supplementary Movies 9, 10 and 11, two ROI are further examined in their dynamics. Whereas the three filopodia (Filo 1, 2, 3) reveal hardly any activity in the 0–10 Hz range, significantly more (actin) activity is visible in both cortex and filopodia on the 10–50 Hz range indicating the necessity of fast sampling. The elongation of filopodia in ROI 2 is accompanied by a tip searching behavior, which becomes visible by zero activity in the central and cortical region of the filopodia (resting parts, read arrows) and the high activity of the filopodia tips (searching parts, green arrows). The underlying driving mechanism for fast lateral tip motions of the actin field filopodia have not been explored so far.

The special capability of ROCS microscopy to observe fast processes over thousands of images is demonstrated in Fig. 3d by adding Latrunculin A, which blocks phosphorylated G-actin and thereby actin polymerization processes, e.g. in filopodia (see non-magnified ROI) and the cell cortex[17]. We added 0.2 μl LatA (dissolved in DMSO) to the cell medium, resulting in a low Lat A concentration of 0.4 nM (see Supplementary Movies 12 and 13). Only 8 s after LatA addition at $t_0$, the cell starts to react strongly: Lat A blocks actin polymerization at the plus-end (see sketches) in the filopodia tips, leading to retractions of actin backbones through retrograde flow, shown by adherent filopodia in the ROIs (Supplementary Movie 15). The 10 ms difference images in grayscale show little dynamics of the adherent filopodia at $t_0 + 0.50$ s, but high tip dynamics only 40 ms later at $t_0 + 0.54$ s. In addition, dorsal filopodia show high dynamics including slightly delayed cell cortex retraction in the difference images (see Supplementary Movie 14), but also in activity maps at Δω = 50-100 Hz (see also Supplementary Note 6). Maximum retraction is reached already at $t_0 + 2.50$ s, going along with a reduced overall activity. Another 45 s later the activity recovers, leading to filopodia growth and cell cortex expansion through massive actin re-polymerization. This behavior is quantified in Fig. 3e: the ROI

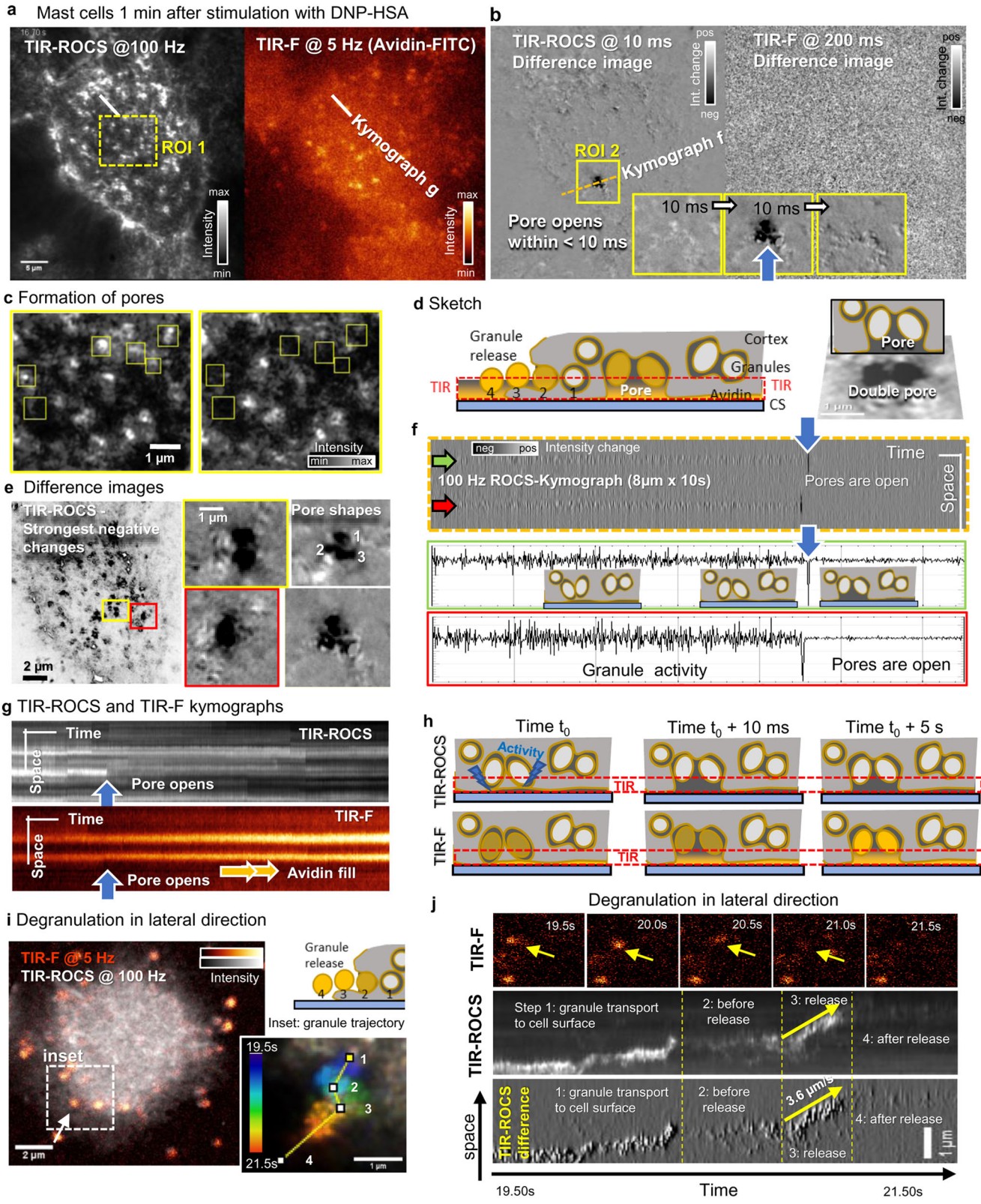

**a** Mast cells 1 min after stimulation with DNP-HSA

TIR-ROCS @100 Hz

TIR-F @ 5 Hz (Avidin-FITC)

ROI 1

Kymograph g

**b** TIR-ROCS @ 10 ms Difference image

TIR-F @ 200 ms Difference image

ROI 2

Kymograph f

Pore opens within < 10 ms

10 ms → 10 ms →

**c** Formation of pores

**d** Sketch

Granule release

Cortex

Granules

TIR 4 3 2 1 Pore Avidin TIR

CS

Pore

Double pore

**e** Difference images

TIR-ROCS - Strongest negative changes

Pore shapes

**f** neg pos Intensity change

100 Hz ROCS-Kymograph (8µm x 10s)

Time

Pores are open

Space

Granule activity

Pores are open

**g** TIR-ROCS and TIR-F kymographs

Time

TIR-ROCS

Space

Pore opens

Time

TIR-F

Space

Pore opens → Avidin fill

**h** Time t₀

Time t₀ + 10 ms

Time t₀ + 5 s

TIR-ROCS

Activity

TIR

TIR-F

TIR

**i** Degranulation in lateral direction

TIR-F @ 5 Hz

TIR-ROCS @ 100 Hz

Intensity

inset

Granule release

4 3 2 1

Inset: granule trajectory

19.5s

21.5s

**j** Degranulation in lateral direction

TIR-F 19.5s 20.0s 20.5s 21.0s 21.5s

TIR-ROCS

Step 1: granule transport to cell surface

2: before release

3: release

4: after release

TIR-ROCS difference

1: granule transport to cell surface

2: before release

3.6 µm/s

3: release

4: after release

space

19.50s Time 21.50s

marked above indicates a radial linescan 1 and a lateral linescan 2 for two kymographs. A nearly exponential decay of the TIR-ROCS signal along a single filopodium reveals a high actin backbone retraction speed of $v = 5\,\mu m/s$. The second kymograph shows the overall signal reduction of 10 adjacent filopodia (previous filopodium marked in red), likely from backbone retractions. A linescan (dark blue) through the whole section of the cell, which

marks the average ROCS intensity decay and recovery within a 60 s time window relative to the signal decay and recovery of the single filopodium marked in red. The magnified time window around the strongest actin retraction at $t_0 + 1$ s, reveals the 40 ms time delay between filopodia and cell cortex. As a control, we added the same amount of DMSO solution without LatA, which led to no visible effects (see Supplementary Movie 16).

**Fig. 4 ROCS microscopy of mast cell degranulation. a** Mouse peritoneal mast cells coated with IgE are triggered by DNP-HSA. 10 s after stimulation the degranulating mast cells are recorded in parallel with TIR-ROCS at 100 Hz and with TIR-F at 5 Hz over 60 s (12 s shown). **b** Difference images reveal significant changes within only 10 ms with ROCS (left), but hardly over 200 ms with Avidin-FITC fluorescence (right). A strong drop in ROCS-intensity (black area in the region of interest ROI 2) suggests a degranulation channel (cavity), which opens within < 10 ms. **c** The magnified ROI 1 shows cell surface areas imaged with TIR-ROCS, which turn into degranulation pores visible as black patches. Further cavities are visible. **d** Sketched cross-section of a mast cell on coverslip within the TIR illumination area. **e** A minimum projection (selection) from 150 ROCS difference images (4.0 s–5.5 s) reveals differently shaped degranulation pores. **f** Kymograph of 1000 ROCS difference images along the line in figure b reveal the opening of two adjacent pores within < 10 ms shown by the black lines (blue arrows). **g** Kymographs along the line scan in figure a: Top: the TIR-F channel shows continuous Avidin-FITC uptake of the vesicles in the pore, the TIR-ROCS reveals a stepwise reduction of intensity from disappearing light-scattering material. **h** cartoon pointing out the possible scenario in the mast cell periphery after stimulus: changes in fluorescence become apparent after decrease of ROCS signals. **i** Degranulation and release of one granule vesicle in lateral direction within TIR-region. Vesicle dynamics is visible both with 5 Hz fluorescence and 100 Hz ROCS imaging. Inset: ROCS image projection over 2 s (200 frames) indicating the subsequent steps 1...4 of granule release. **j** TIR-F signals of degranulation are noisy and become invisible after > 1 s, while TIR-ROCS signals enable detailed kymographs both in intensity and difference images (bottom). Source data are provided as a Source Data file.

**ROCS microscopy of mast cell degranulation**. Mast cells (MC) are key cells of our innate immune response and best known for their roles in allergy, asthma, and anaphylaxis. Activation of mast cells through IgE receptors triggers degranulation with the release of inflammatory mediators that are stored in hundreds of cytoplasmic secretory granules. MC degranulation occurs within few minutes after activation and involves the fusion of secretory granules with the plasma membrane by different modes of exocytosis[18]. Activated mast cells also disassemble cortical actin that favors access of secretory granules to the plasma membrane[19,20]. TIR fluorescence microscopy at 0.15–0.5 Hz rates had previously revealed functional roles for an oscillating actin network[21] and actin depolymerization[19] in promoting granule exocytosis. However, the rapidly occurring steps of granule release at the cell surface are still not fully understood, primarily due to a lack of adequate imaging techniques.

To test the power of ROCS-based imaging for visualizing the fast biological events during mast cell degranulation, we used primary mast cells that had been isolated from the peritoneum of mice and cultured for several weeks.

Figure 4a shows a TIR-ROCS image (at 100 Hz) and a TIR-F image (at 5 Hz, 50 ms integration time) of a mouse peritoneal mast cell, which was coated with an IgE antibody directed against the 2,4-dinitrophenyl hapten (Anti-DNP IgE) and cultured in Tyrodes/FCS buffer with avidin-FITC. Fluorescently labeled avidin recognizes the proteoglycan matrix that is released by exocytosed granules once degranulation is initiated. A region of interest ROI 1 and a linescan are analyzed in Fig. 4c and g. One minute before imaging the degranulation over 60 s with two cameras (Supplementary Movie 17), mast cells were stimulate by adding a DNP-human serum albumin conjugate (DNP-HSA). Figure 4b shows difference images and time intervals of 10 ms (ROCS) and 200 ms (Avidin-FITC fluorescence). Whereas the latter reveals nothing but noise, the TIR-ROCS difference shows slight variations and a pronounced black region marked by ROI 2 and a dashed line for kymograph analysis. Three difference image sequences of ROI 2 reveal the sudden disappearance of ROCS-positive signal patches within < 10 ms (black patch), which suggested the formation of a degranulation pore. Figure 4c shows the same event inside ROI 1 within less than 2 s. Yellow rectangles indicate sudden changes (< 10 ms) of ROCS intensities at the cell periphery. The sketch in Fig. 4d shows a vertical cross-section through a degranulating cell with cavities, which partly open underneath the cell membrane to release granules. After pore formation, the release of granule content can be observed in the lateral direction (steps 1…4). Avidin-FITC binding to the proteoglycan matrix reveals the presence of non-released (hindered) granule content in the TIR region (red dashed area). By a minimum projection (selection) of 150 ROCS difference images, we visualize the strongest negative changes within the time window 4.0–5.5 s in Fig. 4e (see Supplementary Movie 18). Four exemplary shapes of pores are magnified at high image contrast, raising the question whether it can be advantageous that several granules within a single pore are released in parallel—if not hindered by the coverslip.

Figure 4f shows the kymograph of about 1000 ROCS difference images as indicated by the line in Fig. 4b. Two positions of adjacent granules with fast ROCS signal changes over a few seconds are indicated by a green and red arrow. Within 200 ms both pores open (blue arrows, 10 ms broad black lines), leading to a significantly reduced difference signal amplitude, indicated by the two profile plots below. We interpret the fast changes in the difference signals as either granule activity or cytoskeleton reorganization prior to pore opening. Figure 4g displays two kymographs of TIR-ROCS and TIR-F image sequences from the two linescans in Fig. 4a. While the fast pore opening process is visible as a sudden drop of the ROCS signals, this does not occur with avidin-FITC fluorescence. Instead, the avidin-FITC signal continuously increases with time across the double pore, indicating strongly that these pores fill with avidin solution due to direct exposure to the preoteoglycan matrix of released granules. This process is further illustrated by the sketch of Fig. 4h, displaying cell cortex cross-sections at three distinct time points: prior to pore opening, the release of one or several granules is visible by a reorganization of biological material, which instantly stops after pore opening.

Importantly, the combination of TIR-F and TIR-ROCS also allowed the observation of the fast lateral release of granules from the cell surface to the surrounding environment (Fig. 4i). The granules, which have taken up avidin-FITC, are released in several steps 1−4, which can be analyzed much better with TIR-ROCS than with TIR-F (see Supplementary Movie 19). As demonstrated in Fig. 4j, individual granules bleach fast within a few seconds, whereas TIR-ROCS measurements allow the recording of very detailed kymographs from the segmented line in Fig. 3i. Within two seconds (19.5–21.5 s) the granule is transported with fast and significant lateral position changes towards the cell surface (~0.8 s, step 1), rests for ~0.4 s before release (step 2), releases fast over 0.3 s with $v = 3.6$ µm/s (step 3) and moves hardly for the remaining after release (~0.5 s, step 4). Further experiments will be necessary in the future to investigate the spatiotemporal behavior of different granules before and during release with different biochemical conditions established.

**ROCS microscopy of cardiomyocytes and fibroblasts**. Fast cell-cell communication is essential in the cardiac muscle for efficient integration of electrical, mechanical, and humoral signals.

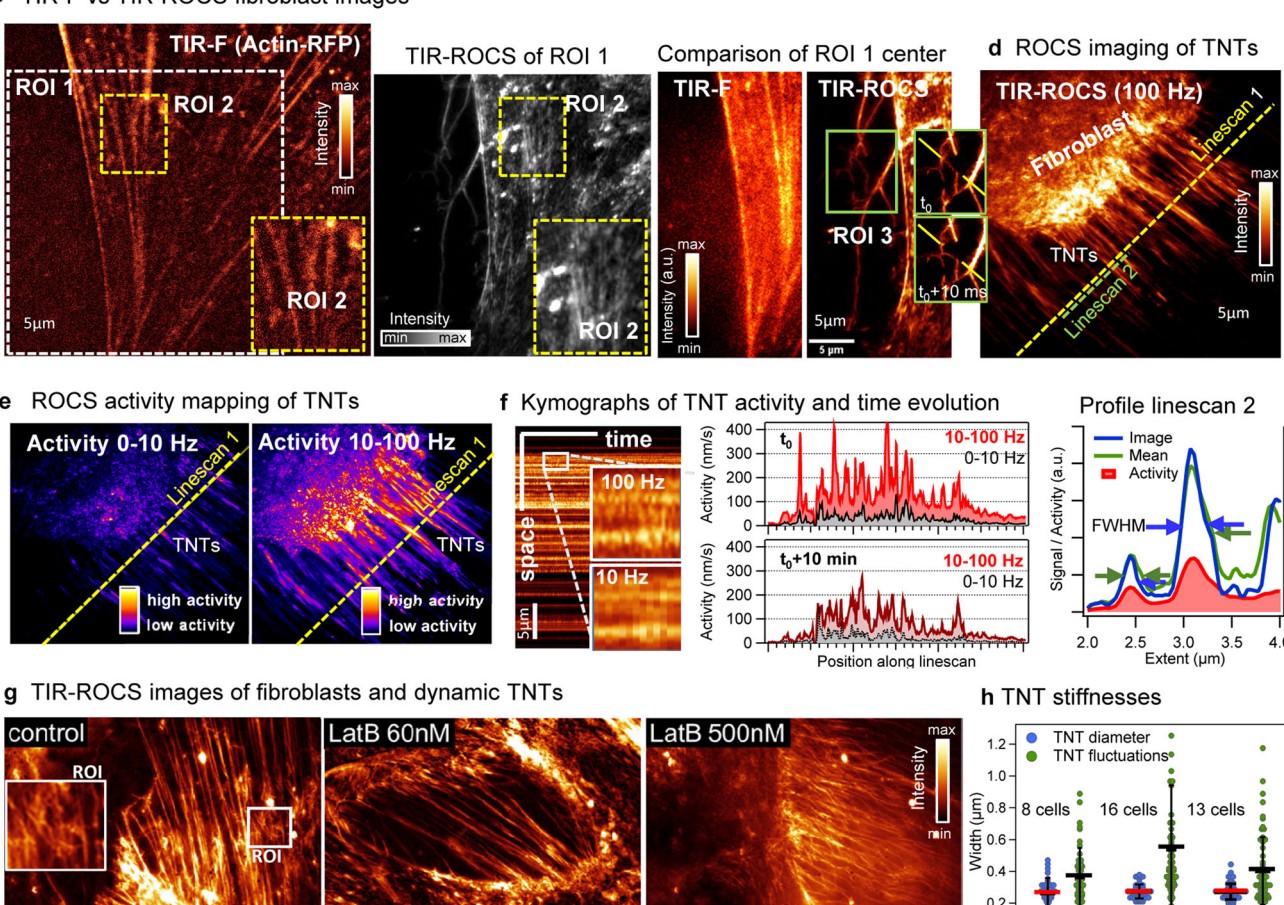

**Fig. 5 ROCS microscopy of cardiomyocytes and fibroblasts. a** Sketch of proposed cardiomyocyte (CM) – fibroblast (FB) interaction through tunneling nanotubes (TNTs) observed by two different polar illumination angles. Red arrows indicate lateral thermal motion of TNTs encoding the TNT stiffness. **b** The FB is imaged with TIR-ROCS, while the CM is imaged with ROCS in non-TIR mode. Image overlay shows enclosure of the CM by FB TNTs. **c** Imaging fibroblasts with TIR-F (actin-RFP) and with TIR-ROCS at 405 nm reveals actin stress fibers with and without fluorescence. TNTs emerging from FB are well visible with TIR-ROCS, but not with TIR-F. **d** TIR-ROCS imaging of FB with many TNTs, linescan 1 (yellow) and linescan 2 (green). **e** FB TNT activity mapped from 100 Hz ROCS movie over 10 s both on slow scale ($\Delta\omega = 0$–10 Hz) and fast scale ($\Delta\omega = 10$–100 Hz) revealing much higher activity on time scales (10–100 ms). **f** TNT dynamics analyzed along linescan 1. Kymographs over 1 s show TNT position changes at 100 Hz and at 10 Hz sampling. Activity profiles $\sigma(x,100\,\mathrm{Hz}, t_0)$ in red and $\sigma(x,10\,\mathrm{Hz}, t_0)$ in gray along linescan 1 (yellow dashed line) at time $t_0$. 10 min later, thermal motion and activity are reduced. Intensity and activity line profiles with FWHM along the Linescan 2. **g** TIR-ROCS images of FBs cultured without and with Latrunculin B (LatB, 24 h). ROI shows cross-connections between TNTs. **h** Exemplary TNT diameters and fluctuation widths σ from 8–16 cells and N = 44–62 TNTs. Horizontal red/black lines indicate the average diameters/fluctuations widths. Source data are provided as a Source Data file.

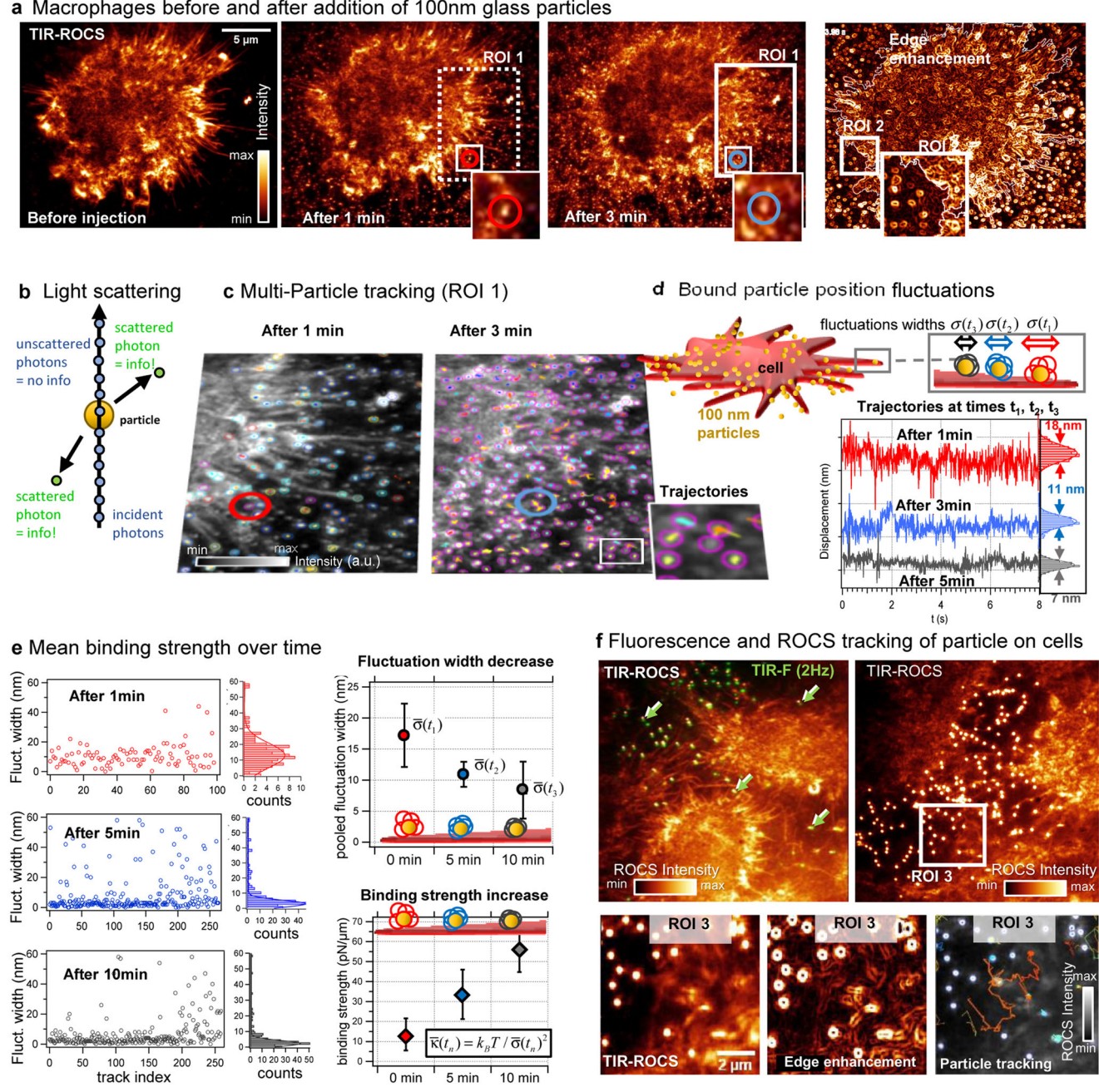

**Fig. 6 ROCS microscopy of particle binding on a thermal noise level. a** TIR-ROCS images (10 ms integration time) of J774 macrophages before exposure of 100 nm glass particles, 1 min after injection and 3 min after injection. ImageJ edge finder increases visibility and high particle densities. **b** A 100 nm glass particle scatters less than 0.6% of incident photons. 0.03 · 0.6% of scattered photons reach the detector carrying information about particle motions. **c** More than 100 particles are tracked automatically within region ROI 1 after 1 and 3 min. A particular particle is marked by a red and blue circle. **d** Position fluctuations of particles bound to the cell surface are analyzed by the position fluctuation standard deviations σ(t). **e** Temporal evolution of mean fluctuation widths $\bar{\sigma}$ and binding strengths $\bar{\kappa}$ of ca. 260 particles after $t = 1$ min, 5 min and 10 min. Within 10 min, the mean binding strength $\bar{\kappa}$ has increased about fivefold. Errors bars indicate STD. **f** Left: image of unlabeled cells with fluorescent-labeled particles recorded with both TIR-ROCS (at 100 Hz) and TIR-F (at 2 Hz). Particles (see 4 white/green arrows) bound to cells are visible with TIR-F (in green, shifted) and with TIR-ROCS (in white/orange). Right: ROCS image of cells with particles. ROI 3 with bright particles in front of cellular structures allowing precise 100 Hz particle tracking. Source data are provided as a Source Data file.

Hetero-cellular coupling between cardiac muscle cells (cardiomyocytes) and interstitial fibroblasts has been shown to be essential both in health and during pathological remodeling[22]. One of the less known modes of cardiomyocyte-fibroblast communication are tunneling nanotubes (TNTs), previously shown to form between the two cell types and proposed to be important for e.g., cardioprotection during ischaemia[23,24].

In contrast to fibroblasts (FB), cardiomyocytes (CM) do not attach to the coverslip, making it difficult to observe them in TIR mode. By decreasing the incident polar angle of the rotating laser beam (the ring diameter in the objective lens BFP), transmitted light can also illuminate and image structures away from the coverslip (non-TIR mode). This principle is sketched in Fig. 5a, where the interaction of rabbit primary left ventricular CM and

primary human atrial FB-borne TNTs can be observed in two separate images obtained within 20 ms by 100 Hz switching between two distinct polar angles. The red arrows indicate lateral thermal motions of the TNTs, where the fluctuation amplitude encodes the stiffness of the tubular inter-cell connection at a specific time. Two-angle ROCS imaging is shown in Fig. 5b, where the superposition of both images displays how the FB TNTs enclose the CM. The thin TNTs scatter significantly less light than the surface of the CM, thus compromising the quality of the CM image only slightly.

Figure 5c demonstrates the visibility of single filopodia-like cell protrusions such as TNTs, which are difficult to discern in TIR-F mode, but clearly resolved in TIR-ROCS mode. The FB labeled with actin-RFP (excited at 561 nm) exhibits stress fibers which can be seen also in ROCS, although neighboring structures scatter as well (see ROI 2). ROI 3 shows a branching protrusion where a filament flips within only 10 ms as indicated by the yellow lines.

Figure 5d displays a snapshot from a 100 Hz image sequence over 10 s (see Supplementary Movie 20) of a FB with numerous TNTs. The TNTs undergo specific motions, which were analyzed along the yellow and green dashed linescans. Figure 5e shows the corresponding 2D activity maps for motions in the frequency range $\Delta\omega = 0$–$10$ Hz and in the range $\Delta\omega = 10$–$100$ Hz, where the latter indicates fast motions, which should be thermally driven, but not actively by the cell on these timescales (see Supplementary Movie 21). A kymograph along linescan 1 with two magnified ROIs illustrates how TNTs fluctuate in position on the millisecond timescale, which is hardly visible with a 10 Hz movie rate (Fig. 5f). This is further analyzed by the ROCS activity (velocity standard deviation 0–400 nm/s) of the TNT at two specific time points. Notable are two things: first, activity is hardly visible at a 10 Hz acquisition rate (gray profiles), and second, the activity of the TNTs decreases significantly within a 10 min time window, resulting from decreasing fluctuation amplitudes, possibly due to stiffening of the TNTs. This interesting observation related to FB-CM interaction will be investigated in much more detail in the future. The motions of three adjacent TNTs along linescan 2 (in Fig. 5d) are further illustrated by the full-width at half maxima (FWHM) of a single 10 ms ROCS signal (in blue), of the slightly broader averaged ROCS signal (in green), and the activity profile (in red), which is the most robust measure for motions at high sampling rates.

Figure 5g, h provide an outlook towards future experiments to investigate TNT formation and stiffening over time, for example upon exposure to different cytoskeleton-disrupting substances (such as Latrunculin B) as displayed in Fig. 5h.

**ROCS microscopy of virus-sized particle binding on a thermal noise level.** Small, submicrometer sized particles, such as airborne particulates (particulate matter, PM) or viruses dispersed in micro-droplets cause lung infections leading to severe respiratory diseases. The disease COVID-19 spreads because of amplified exchange of 100–120 nm small replicating coronaviruses entering lung cells[25]. Diseases such as asthma, lung cancer, cardiovascular disease, and stroke[26], are a likely consequence of worldwide combustion processes and air pollution. Both coronaviruses and ultrafine particulates ($PM_{0.1}$, diameters < 300 nm) can diffuse down the lung to the alveols, enter and damage epithelial cells (and macrophages), hence preventing sufficient air exchange. However, because of a lack of sufficiently fast and robust microscopy techniques the dynamics of such small particles and their binding behavior to the cells are widely unexplored.

Using 100 Hz TIR-ROCS, we have observed the dynamics of 100 nm glass beads and their binding behavior to J774 macrophages (see Supplementary Movies 22 and 23). These

beads have a similar size and refractive index as coronaviruses and some ultrafine particulates[27–29]. As shown before, a too low temporal resolution leads to motion blur and complete invisibility of the highly dynamic particles with 600 ms mean free diffusion length within 100 ms and in a solution 5x more viscous than water. Figure 6a displays the 10 ms ROCS image of a cell before the injection of the 100 nm particles. The next two images show the in-diffusion of particles after 1 min and 3 min, pointing out ROI 1 and one individual particle enclosed by a red and blue circle. Even without edge enhancements in the fourth image, the particles can be well observed without significant motion blur (with lengthy particle images) as indicated by ROI 2.

Figure 6b illustrates that a $2a = 100$ nm glass particle scatters less than $Q_{sca} = 0.6\%$ of the incident photons ($Q_{sca} = C_{sca}/a^2\pi$ based on Mie theory). For a NA = 1.2 lens 30% of the scattered light is captured, and with a 10% overall transmission to the camera, only $0.3\cdot0.1\cdot0.6\% = 1/5555$ of scattered photons reach the detector carrying information about particle motions. Due to coherent amplification, this was sufficient to precisely track >100 particles at 100 Hz inside region ROI 1 (e.g. by using TrackMate[30]) as displayed in Fig. 6c and Supplementary Movie 24. The fate of the single-particle enclosed by the (red/blue) circle is further investigated in Fig. 6d, where trajectories were analyzed over 8 sec (800 positions) at three different time points after cell binding. The thermally driven position fluctuations reduce from $\sigma = 18$ nm to 11 nm to 7 nm after $t_1 = 1$ min, $t_2 = 3$ min and $t_3 = 5$ min indicating a tighter binding to the cell. $\bar{\sigma}$ is the 61% standard deviation of the position histogram obtained from the trajectories. As shown by Fig. 6e, we estimated the mean particle binding strength $\bar{\kappa}(t) = k_B T/\bar{\sigma}(t)^2$ via the equipartition theorem for totally 260 particles after 1 min, 5 min, and 10 min. Approximately, the fluctuation width is reduced by a factor of 2, whereas the membrane-binding strength increased nearly 5 fold, possibly because of integrins in the cell[31].

To prove that we can successfully image such small particles in front of cells, we acquired images of the unlabeled cells with fluorescently labeled 120 nm PS particles with TIR-ROCS (at 100 Hz) and TIR-F (at 2 Hz) as demonstrated in Fig. 6f and Supplementary Movie 25. Unbound and bound particles are well visible with TIR-F (in green, shifted) and with TIR-ROCS (in white/orange). The magnified region ROI 3 reveals bright particles in front of cellular structures allowing precise 100 Hz particle tracking, as illustrated by the grayscale figure with the orange trajectory.

TIR-ROCS imaging allows to investigate the thermally driven binding behavior of small particles to cells with standard tracking procedures and with high statistics, allowing to perform various binding tests in the future.

**ROCS microscopy of LecA dynamics at the cell cortex.** Bacteria produce lectin proteins in order to mediate adhesion and entry into host cells by binding to glycosylated plasma membrane receptors. The interactions between the *Pseudomonas aeruginosa* lectin LecA and its host cell glycosphingolipid receptor Gb3 (globotriaosylceramide) trigger the cellular uptake of the bacterium via the so-called lipid zipper mechanism[32,33]. However, the dynamics of LecA at the plasma membrane leading to membrane invaginations and the mechanisms of clathrin-independent endocytosis remain so far largely unknown.

We investigated the interactions of LecA with the periphery of H1299 lung epithelial cells with two cameras as shown in overlay image of Fig. 7a. LecA clusters labeled with Cy5 were imaged in TIR-F mode at 2 Hz (50 ms integration time) parallel to 100 Hz TIR-ROCS, revealing both LecA motions and different cortex dynamics 20 min after addition of LecA (5 µl/ml), as shown in

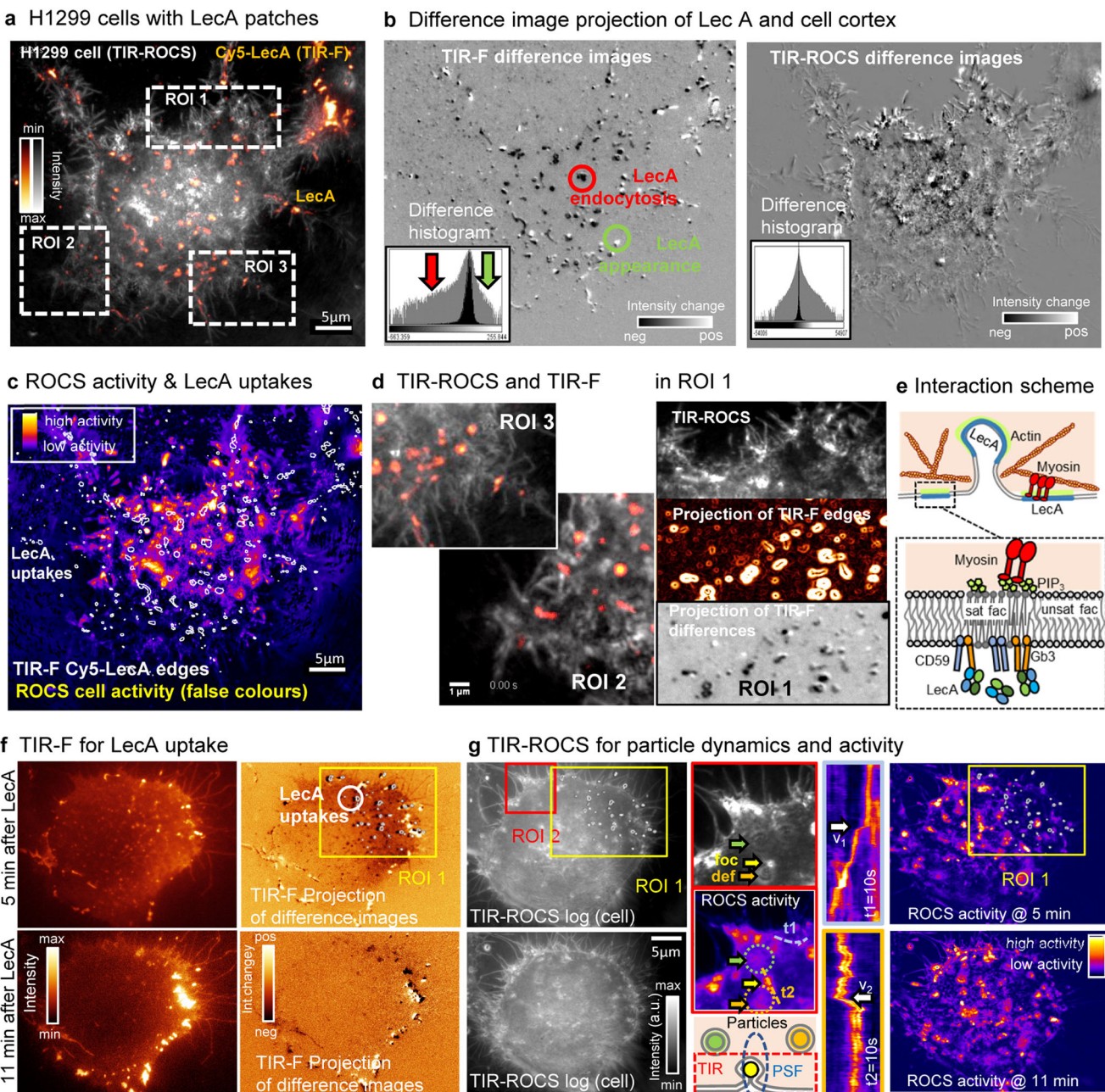

**Fig. 7 ROCS microscopy of LecA dynamics at the outer cell cortex. a** TIR-ROCS image (10 ms integration time) of a H1299 lung carcinoma cell in gray colors overlaid with TIR- image (50 ms integration time) of LecA clusters (Cy5) in orange colors. **b** Difference images averaged over 10 s show dynamics of LecA clusters (TIR-F) and (TIR-ROCS). The difference histogram is asymmetric, negative biased for TIR-F indicating enhanced endocytosis. Negative (positive) differences in black (white) correspond to endocytosis (exocytosis). **c** TIR-ROCS activity map over 10 s and full 100 Hz bandwidth show cell cortex dynamics relative to edges of LecA TIR-F images. **d** Regions of interest ROI 3, ROI 2 (see Supplementary Movies) and ROI 1 from figure above show cell cortex details (ROCS), the occurrences of Cy5-Lec A clusters (edge enhanced) and the LecA dynamics by TIR-F difference averaged over 10 s. **e** Sketch of membrane invagination and transport of LecA along actin fibers through myosins. Outside the membrane LecA binds to Gb3 and CD59, which loosely connects through the membrane to long saturated fatty acid chains of PIP3 and myosin. **f** Imaging of LecA with 2 Hz TIR-F after 5 min and 11 min of LecA addition. TIR-F differences averaged over 10 s reveal several uptake events (black patches and corresponding white edges, right-shifted). **g** TIR-ROCS imaging of H1299 cell after 5/11 min of LecA addition with white edges in ROI 1 from LecA uptake areas in f. ROI 2 shows one (foc)ussed particle (yellow arrow) at membrane and two (def)ocussed particles inside the cell, which cause activity as well (dashed circles in activity ROI below). The kymographs along t1 and t2 depict stop-and-go diffusion with maximal velocities $v_1$ and $v_2$ (white arrows). Activity plots at 5/11 min show high particle dynamics at the membrane and in the cortex. Source data are provided as a Source Data file.

Supplementary Movies 26,27 and 28. After bleach correction, the overall dynamics of LecA were analyzed by difference images (Supplementary Movie 30), which we averaged over 10 s (Supplementary Movies 29). Figure 7b reveals significantly more

black patches (red circle) than bright patches (green circle), indicating endocytic uptake of LecA, which is also confirmed by the asymmetric histogram (see red and green arrows). However, we could not find any correlation between LecA dynamics and

the cell cortex dynamics - neither had we found many black patches in ROCS difference images, nor asymmetry histograms. A full bandwidth ROCS activity map overlaid with the edges of LecA clusters (Fig. 7c) displays no enhanced cortex activity around LecA, but significant motion of particles in the cell membrane or below. The regions ROI 3, ROI 2 (see Supplementary Movie 27) and ROI 1 in Fig. 7d illustrate the high image quality of parallel TIR-ROCS and TIR-F imaging.

Based on a current working model[34], LecA binding induces a plasma membrane domain enriched in saturated Gb3 species and the GPI-anchored protein CD59 at the extracellular membrane leaflet, which probably is able to induce membrane curvature required for endocytic uptake. We hypothesize that trans-bilayer signal propagation could be achieved by long fatty acyl chains of Gb3 and CD59 that interdigitate into the cytosolic membrane leaflet, leading to interactions and clustering of long fatty acyl chains of the phospholipid PIP3, to which myosins as actin motor proteins can bind.

We investigated the correlations between LecA uptake with TIR-F (Fig. 7f) and the cell cortex activity with TIR-ROCS (Fig. 7g) over 25 min. Five minutes after addition of LecA the TIR-F image displays plenty of Cy5 fluorescent patches, many of which disappeared within the following six minutes. Continuous uptake of LecA within 17 s is revealed by black patches (also indicated by white edges) in the projected TIR-F difference, but not in the projected TIR-ROCS difference. After 11 min LecA uptake has decreased significantly. The logarithmic TIR-ROCS image as part of Fig. 7g and Supplementary Movie 31, reveals no correlation to the signals in the activity maps at 5 min or 11 min, in particular not in the uptake areas indicated by ROI 1. However, high dynamics of particles on the membrane (possibly LecA clusters, e.g., at yellow arrow) and defocused particles below the membrane (possibly endosomes after LecA uptake and vesicle fusion, e.g., at green/orange arrow) are visible as shown in ROI 2 (and Supplementary Movie 31). Exemplary kymographs over 10 s along trajectories t1 and t2 reveal hindered and driven diffusion of LecA clusters), possibly due to inhomogeneities of the actin cortex (and myosin transport with maximum velocities of $v_1 = 4.0 \, \mu m/s$ and $v_2 = 4.3 \, \mu m/s$.

## Discussion

We have shown that ROCS microscopy provides thousands of images at a very high spatiotemporal bandwidth (STBW) of $100 \, Hz/0.16 \, \mu m = 625 \, Hz/\mu m$ without labeling and without loss in image quality. Comparable STBWs can be achieved with fluorescence microscopy (TIRF-SIM)[3] only by using a priori information (e.g., filamentous shapes). ROCS gives researchers the possibility to observe the decisive seconds or minutes of interaction, either starting spontaneously or induced, at an unknown point in time. We showed that the combination of ROCS in TIR mode and darkfield mode achieves extraordinary image contrast of smallest dynamic structures within only 10 ms - an integration time which has not been minimized yet, e.g. with better cameras.

Motion blur limits resolution. We have demonstrated at several examples that limited temporal resolution can be more significant than limited spatial resolution. Motion blur can turn objects completely invisible, when their photon emission power $P_{em}$ is distributed over a too large motion area A, such that the intensity $I_{em} = P_{em}/A$ approaches the background level. Unexpectedly, we found that 100 Hz acquisition is still not enough for several experiments, (which generates typically 4 GB of raw data in 10 s). Based on the diffusion estimates presented in the introduction, a 2–5 fold faster image acquisition rate would be more beneficial, requiring faster scan mirrors and cameras. As shown in the

Supplementary Note 7, it is important to check that the cell activity at the beginning and the end of the experiment does not change, to exclude obvious phototoxic cell damages, although only 1 mW laser power reaches the sample (illumination I = 0.3 W/mm²). In this context, we also investigated the effects of different laser wavelengths (405 nm or 445 nm) on the vitality of cells.

Coherence controls speckles and resolution. Object illumination and detection of laser light achieves about tenfold higher photon densities because of interference. Based on highly oblique illumination, the object spectrum is shifted in ROCS, such that significantly more information is transferred close to the cutoff frequency (see Fig. 1e and Supplementary Fig. S3). Furthermore, the high degree of (radial) spatial and temporal coherence leads to destructive interferences between two objects in distances $d < \lambda$ and thereby further increases resolution. For single direction illuminations, the multiple interferences can lead to static, deterministic speckles, which are ill-formed images of object structures. These speckles turn into well-formed image structures after full $2\pi$ angular ROCS-illumination. However, even in full $2\pi$ ROCS microscopy the object images can turn into speckles or into ill-formed image structures, if the object structure sizes come below the current resolution of about 160 nm. This resolution limit can be further pushed in the future by applying deconvolution algorithms (as e.g. in SIM), by deep learning[35] and by increasing the effective numerical aperture of currently $NA_{det} = 1.2$ by blocking the reflected rotating laser focus only locally.

What do we really see with ROCS? At the camera, we see a superposition, i.e. interferences of elementary waves, which vary in phase and amplitude according to the complex refractive index and polarizability of the cellular structures. This gives room for future improvements of interference patterns, which depend on the polar angle of incidence and the laser wavelength. In contrast to the related technique of Ptychography[36,37], ROCS needs no postprocessing and illuminates the sample only at 1–2 high polar angles (laser rotation at high NA) to achieve fast image acquisition. And, by using fully coherent light, spatial resolution and contrast can be maximized[8]. By imaging cells in TIR mode, only the outer cortex is illuminated in a depth of 0.2–0.3 μm. As demonstrated in the experiment of Fig. 2c with laser tweezers moving and exerting forces in the outer cell cortex, ROCS signal changes could be induced. Because of the ring-like illumination in ROCS, high spatial frequencies are boosted leading to edge enhancement. Hence, cell structures such as flat areas in lamellipodia remain invisible with ROCS, when the refractive remains constant. Spherical vesicles generate clear ROCS signals, such that most likely actin is responsible for the observed dynamics and the activity signals, which we map on different frequency bands. Actin stress fibers can be seen with TIR-ROCS (see Fig. 5c), individual actin structures as well, if the cortex density is not too high (Figs. 1g and 3a). ROCS imaging provides a useful signal control for fluorescence images depending on the cell's fluorescing protein expression level (see Supplementary Note 8).

Activity maps summarize an image sequence. We have demonstrated that single and averaged difference images can reveal more structural changes than conventional analysis of image intensities. Due to the high ROCS image contrast, ROCS difference images hardly show any noise. A whole movie sequences can be analyzed and summarized in a single image only, called activity map[15], which represents the standard deviation of several difference images. These maps are advantageous, especially when decomposed into different frequency ranges (typically $\Delta\omega = 0–1 \, Hz$, $10–33 \, Hz$, and $33–100 \, Hz$). This becomes apparent, when a structure moves on both a low-frequency and a high-frequency: the slow motion can cover the fast motion, which can be uncovered by this decomposition (see Figs. 3c and 5e).

Filopodia tips undergo stochastic searches. By several TIR-ROCS image sequences with corresponding activity maps (and 3 Supplementary Movies), we investigated the behavior of elongating filopodia of J774 macrophages at 100 Hz over 5000 images. With exceptional spatiotemporal resolution we could show, how single, mainly adherent filopodia grow in length while the tip region (of ≈1 μm length) fluctuates strongly in lateral directions. These motions can be thermally driven, but also by a yet unknown mechanism. Nevertheless, the fast direction changes do not inhibit actin polymerization at the tip, but apparently impede adhesion to the surface. Activity plots reveal that fluctuations are more pronounced on a 10–30 ms scale than on 0.1–1 s timescale.

Cell response to Latrunculin A within a few seconds. Our goal was to record fast cellular responses at unknown time points upon biochemical stimuli. We demonstrated that polymerization of actin fibers in the cortex and of filopodia actin backbones was hindered, by blocking ATP-G-actin within a few seconds after adding 0.4 nM Latrunculin A. This was visualized by TIR-ROCS movies, difference images and high-frequency activity maps ($\Delta\omega = 50$–$100$ Hz). ROCS revealed visible retractions not of whole filopodia, but only of the actin backbone at high velocities of up to $v_{BB} = 5$ μm/s. Since off-binding of (ADP) actin is on the order of 10 Hz at the barbed end[38], this results in a maximal depolymerization speed of only 5.4 nm/2·10 Hz = 27 nm/sec $\ll v_{BB}$. Retraction velocities of actin backbones in J774 macrophages, based on myosin II motor pulling, have been reported to be 0.1-0.3 μm/s[39] with LatA addition. The whole cortex retracted a few 100 ms later and over several μm within only 1–2 s, followed by a minimum of activity after a few seconds. Then, the cell cortex activity was recovered within 60 s through fast re-polymerization of unblocked G-Actin going along with an expansion of the cell. The observed effects of actin inhibition by LatA were significantly faster than that those reported on fluorescence-labeled actin[17] and open new questions about mechanical cytoskeleton retraction mechanisms and the efficiency of myosin—actin interaction (especially of unlabeled, wild type f-actin).

Mast cell degranulation pores form within a few milliseconds. Given the lack of adequate fast imaging techniques, the spatiotemporal sequence of fast occurring events that underlie MC degranulation has not yet been resolved in detail. Using 100 Hz TIR-ROCS, we could see the formation of degranulation pores in stimulated mast cells 1 min after IgE-mediated activation. By difference images, we could see pore formation within <10 ms in exceptional temporal resolution, revealing clear shapes and boundaries, indicating the number n of granules to be released (typically $n = 2$–$5$). While granule release was frustrated at the cell bottom, we could observe the lateral release of individual granules in multiple steps, which could be analyzed quantitatively through ROCS kymographs. Further investigations with even higher acquisition rates may provide further insights into the driving mechanisms of ultrafast pore opening and granule release.

Nanotubes of cardiac fibroblasts fluctuate on ms scales. Whereas fibroblasts close to the coverslip where imaged with TIR-ROCS, cardiomyocytes further away could be observed 10 ms later with non-TIR-ROCS. We then recorded repetitive image sequences of long tunneling nanotubes (TNT) protruding from fibroblasts, which were difficult to visualize with fluorescence. Using 100 Hz TIR-ROCS, we measured lateral TNT fluctuations likely driven by thermal noise. Ten minutes later and at only 9% reduced cell cortex activity, the TNT motions were significantly reduced in amplitude and time, likely because of the stiffening of the TNTs by yet unknown mechanisms. Changes in TNT stiffness might be an important determinant of cardiomyocytes-fibroblast cross-talk in the heart during homeo-stasis and after injuries.

Thermal motions of virus-sized particles reveal binding behavior. We used 100 nm small glass particles, mimicking ultrafine particulates or viruses, to record their spatiotemporal binding behavior to macrophages. Due to the high ROCS image contrast, automated tracking of hundreds of particles allowed to analyze the fluctuation behavior at different stages prior and during binding to the filopodia-rich cell periphery. We found a fivefold increase in binding stiffness within only 10 min, indicating structural changes at the cell periphery. In the future, these processes must be investigated in more detail by adding receptor blockers or different molecular particle coatings. The extremely fast hopping of tiny particles between potential binding sites indicates the complexity of particle cell interactions.

LecA is driven by unknown mechanisms. We imaged Cy5 labeled LecA diffusing at the surface of HT1299 cells and below in a combination of TIR-F and TIR-ROCs. Interestingly, no correlations between the Cy5 fluorescing LecA dynamics and highly dynamic refractive index changes within the cell cortex could be found. While LecA positions and uptake positions were clearly located by difference images, no locally enhanced actin cortex activity could be revealed by ROCS. However, it turned out that a combination of hindered and driven diffusion of membrane LecA patches with velocities up to 4 μm/s could be tracked precisely, as well as dynamic, defocused particles below the membrane (likely LecA-containing endosomes), which could be explained by a reduced density of the actin cortex and active transport by myosins binding to PIP3. For sure these observations open interesting questions about the driving mechanisms of both LecA clusters and endosomes, which should be investigated in the future by multiple-plane ROCS microscopy.

We could show by several examples of cellular systems that 100 Hz ROCS microscopy reveals various unexpected findings in the high-speed world of molecular cell structures, which in some cases would require even faster measurements. In combination with the specificity provided by fluorescence techniques, ROCS represents a profitable imaging approach, which can be well added to existing microscopes. By extracting molecular absorption and specificity through phase delays[40], by achieving optical sectioning through e.g., speckle correlations and addressing 3D imaging through lightsheet illumination or multiple-plane detection, we envision a bright and promising future for ROCS microscopy not only for cell biology.

## Methods

**Ethical statement**. We confirm that our research complies with all relevant ethical regulations; approved by the local Institutional Animal Care and Use Committees in Germany (Regierungspraesidium Freiburg, X-16/10 R).

We declare that no wild animal were used in the study and that no field-collected samples were used in the study.

**Cell culture methods**. J774 cells (ATCC TIB-67, www.atcc.org/cell-products) transfected with pLife Act-TagGFP2 (ibidi GmbH, Germany) were cultured in DMEM medium and in a 37 °C and 5% CO2 atmosphere until the desired cell density for the experiments is reached (approx. 30% confluency). This cell line was tested negative for mycoplasma. For bead experiments, beads were pipetted into the cell culture medium and the imaging process was started with a controlled time delay as stated in the manuscript and figure legends. For the experiments with Latrunculin A, LatA was dissolved in DMSO at a concentration of 200 nM. 2 μl of LatA-DMSO solution was added to the cell medium (1 ml), resulting in a low Lat A concentration of 0.4 nM. Then recordings were started immediately.

Primary mouse immune cells were isolated from C57BL/6 J mice (The Jackson Laboratory). Mouse breeding and husbandry were performed at the Max Planck Institute of Immunobiology and Epigenetics, Freiburg, in accordance with the guidelines provided by the Federation of European Laboratory Animal Science Association and as approved by German authorities (Regional Council of Freiburg). Mice were kept at a light/dark cycle of 14/10 h, 22 ± 2 °C temperature and 60 ± 5% relative humidity. C57BL/6 mice were only used for organ removal after euthanasia by carbon dioxide exposure and thus not subject to experimental procedures and ethical approval according to §4 (3) Tierschutzgesetz. Mouse neutrophils were isolated from bone marrow using autoMACS Pro Selector cell separator and MACS neutrophil negative selection kit according to the manufacturer's protocol (MIltenyi Biotec). Mouse peritoneal mast cells were

isolated by peritoneal lavage and cultured before use in OptiMEM Medium supplemented with 10% FSC, GlutaMAX, penicillin, and streptomycin[41].

Mast cells were seeded with approx. $35 \times 10^4$ cells per dish in 2 ml culture medium. Cells were coated with an IgE antibody directed against the 2,4-dinitrophenyl hapten (Anti-DNP IgE, 0.5 µg/mL; Sigma-Aldrich, D8406) overnight, before mast cell degranulation was induced with a DNP-human serum albumin conjugate (DNP-HSA, 100 ng/mL; Sigma-Aldrich, A6661). Cells were kept in Tyrodes/FCS buffer that was supplemented with FITC-conjugated avidin (0.5 µg/mL; Biolegend, 405101), which recognizes the proteoglycan matrix released by exocytosed granules. Mast cell degranulation at the given concentration was validated by FACS measurements for avidin and CD63. Further references on manufacturer's website: https://www.sigmaaldrich.com/DE/de/product/sigma/d8406

Two types of cardiac cells (human primary atrial fibroblasts, rabbit primary left ventricular cardiomyocytes isolated using Langendorff-based enzymatic dissociation) were used. All investigations reported here conformed to German (TierSchG and TierSchVersV) animal welfare laws, compatible with the guidelines stated in Directive 2010/63/EU of the European Parliament on the protection of animals used for scientific purposes, and they were approved by the local Institutional Animal Care and Use Committees in Germany (Regierungspräsidium Freiburg, X-16/10 R). Animal housing and handling were conducted in accordance with good animal practice, as defined by the Federation of European Laboratory Animal Science Association, FELASA. Rabbit tissue was obtained from New Zealand white rabbits (1 yo, mixed sex). Human tissue samples were obtained from the right atrial appendage of the patient undergoing open-heart surgery at the University Heart Center Freiburg - Bad Krozingen. Informed consent was acquired prior to surgery. Tissue samples were processed by the Cardiovascular Biobank of the University Heart Center Freiburg - Bad Krozingen (approved by the ethics committee of Freiburg University, No 393/16; 214/18). Cells were maintained in standard culture conditions prior and throughout the experiments (DMEM GlutaMAX, supplemented with 10% FCS, 1% penicillin/streptomycin). Cultures containing CM were maintained for maximum of 3 days. Latrunculin B (LatB) treatment was applied 24 h prior to imaging. Control was exposed to vehicle (ethanol) alone. Actin was visualized using CellLight™ Actin-RFP, BacMam 2.0 (Thermo Fisher).

Mycoplasma-free H1299 cells (ATCC CRL-5803, www.atcc.org/cell-products) were cultured in RPMI medium and in a 37 °C and 5% CO2 atmosphere until the desired cell density for the experiments was reached (approx. 30% confluency). Cells were gently washed once in PBS prior to lectin addition. Cy5-labelled LecA was added with a concentration of 5 µg/ml. Excitation of LecA was done with a 561 nm laser (Cobolt) operating at 4 mW at the sample. The exposure time was 50 ms unless stated otherwise for the best compromise between photon signal and motion blur.

**ROCS imaging**. ROCS imaging was performed at 100 Hz under TIR condition with a laser power of < 1 mW at the sample, unless stated otherwise. Every optical surface/interface, especially between and sample and camera, must be anti-reflection-coated for the specific laser wavelength and must be completely free of dust/dirt, which introduces unnecessary image artifacts. Further details can be found in the Supplementary Note 1.

**Reporting Summary**. Further information on research design is available in the Nature Research Reporting Summary linked to this article.

## Data availability
Source data are provided with this paper in the Source Data file. Image data supporting the main figures in this work are available at https://doi.org/10.5281/zenodo.6122535. Further information regarding design and cell preparations may be found in the Nature Research Reporting Summary. Source data are provided with this paper.

## Code availability
Analysis codes are available at https://doi.org/10.5281/zenodo.6122535.

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

## Acknowledgements

The authors thank Carsten Schwan and Roland Nitschke for helpful discussions and comments on the manuscript. This study was supported by the German Research Foundation (grant RO 3615/14-1, AR) and the German Collaborative Research Centre SFB 1425 (DFG 422681845, AR, ERZ), by the Excellence Initiative of the German Federal and State Governments (EXC 294 and EXC 2189, AR and WR) and by the German Research Foundation grant 278002225 (RTG 2202 - Transport across and into membranes, WR) paying Annette Brandel. Rebecca Michiels was funded by the German Research Foundation (grant RO 3615/3-2, AR). Alina Gavrilov, Michael Mihlan, and Tim Lämmermann were funded by the Max Planck Society. We thank the Freiburg Cardiovascular Biobank (CVBB) for access to human tissue and SCI-MED facility for hardware excess. We also thank Dr. Nicole Gensch from the Freiburg BIOSS Toolbox for Transfection of the J774 cells.

## Author contributions

A.R. initiated the project. F.J., D.R., and A.R. designed the ROCS setup. F.J. performed most experiments, D.R. measured results of Fig. 3a, D.S. of Fig. 2a, D.H. of Supplementary Fig. S2. R.M., F.J., A.B., A.G., M.M. and C.D. prepared the various cells and experiments. E.R.Z., T.L., J.M. and W.R. helped to design the experiments and to discuss results. F.J. and A.R. analyzed data and prepared figures. A.R. wrote the manuscript.

## Funding

## Competing interests

The authors declare no competing interests
