## [Peer Review File · Nature Communications]

Reviewers' Comments:

Reviewer #1:

Remarks to the Author:

The manuscript applies Rotating Coherent Scattering (ROCS) microscopy to a wide range of systems, demonstrating its high temporal resolution at diffraction-limited spatial resolution. The core advantages of ROCS microscopy are its high imaging speed, good sensitivity and resolution, while not requiring fluorescent labeling. Thus, ROCS will be important for all biophysical/biological applications where one needs high imaging speed of extended periods of time, where fluorescence methods will fail due to signal intensity limitations and photo-bleaching.

I have carefully read all reviewers' comments and the rebuttal and revised version by Rohrbach et al. In my view, the authors have addressed all the raised technical question adequately.

Beyond the technical questions, there have been raised general objections:

Both reviewers questions the novelty of the manuscript. Indeed, the working principle of ROCS and several applications of it have been published before, but the core advance of the current study is to demonstrate its capability for a wide range of diverse systems and conditions, which makes this paper a landmark study of ROCS and an excellent presentation of all its capabilities (and limitations). Thus, I indeed think that the paper is novel and substantial enough to warrant publication in *Nature Biotechnology* - never before has the method of ROCS been presented in such a broad scope, with so many details of the measurement technique and data analysis in one place.

Reviewer 3 states: "... most of the presented biological studies showed that ROCS is complementary to TIR-F." Sure, in terms of resolution yes, in terms of speed sure not. When imaging fluorescent samples with 1.5 kHz, the signal strength will be in most cases unacceptably low, or one needs extreme label densities. Moreover, the required excitation intensity would bleach a sample within shortest time. Thus, I think, to claim that TIR-F is equivalent to ROCS is unfair.

A more serious objection is that there exists other coherent label-free microscopy techniques such as iSCAT, GLIM, or Optical diffraction tomography. This is certainly true, and one has to see where ROCS will figure among all these techniques. But I think as a promising alternative technique and with its many successful applications as shown in the manuscript, it indeed merits to be published in *Nature Biotechnology*.